# Where to draw the line?

Heping Sheng[1], John Wilder[2], Dirk B. Walther[2]*

1 School of Medicine, Boston University, Boston, MA, United States of America, 2 Department of Psychology, University of Toronto, Toronto, Canada

* bernhardt-walther@psych.utoronto.ca

## Abstract

We often take people's ability to understand and produce line drawings for granted. But where should we draw lines, and why? We address psychological principles that underlie efficient representations of complex information in line drawings. First, 58 participants with varying degree of artistic experience produced multiple drawings of a small set of scenes by tracing contours on a digital tablet. Second, 37 independent observers ranked the drawings by how representative they are of the original photograph. Matching contours between drawings of the same scene revealed that the most consistently drawn contours tend to be drawn earlier. We generated half-images with the most- versus least-consistently drawn contours and asked 25 observers categorize the quickly presented scenes. Observers performed significantly better for the most compared to the least consistent half-images. The most consistently drawn contours were more likely to depict occlusion boundaries, whereas the least consistently drawn contours frequently depicted surface normals.

**Data Availability Statement:** All data and data analysis scripts as well as all individual line drawings and the super-references are available on the Open Science Framework website: https://osf.io/x9uj5/.

## Introduction

Humans have used line drawings to depict scenes from their lives for at least 45,500 years [1]. Line drawings can capture the essence of shapes and spatial relationships in complex scenes to an extent that makes them not only reasonable stand-ins for photorealistic images, but can emphasize particular aspects of the scene unencumbered by other, extraneous features that might otherwise obscure the clarity arising from a purely contour-driven depiction. For this reason, line drawings are used for technical drawings, such as architectural plans, designs for complex machinery, or charts for complex production processes. Even though line drawings lack texture and color, they nevertheless represent essential information, making them suitable for efficiently conveying visual information non-verbally, for instance in signs that request patrons to wear masks inside stores, instructions for the Heimlich maneuver, or step-by-step instructions for assembling a MALM chest of drawers from IKEA.

Line drawings of real-world scenes are recognized as quickly and accurately as photographs [2, 3]. Perception of line drawings is innate to human children [4–6] and does not rely on an acquired cultural framework [7, 8]. Even chimpanzees were found to understand line drawings without prior training [9]. In their first attempts to artistically reflect their world, children resort to line drawings [10]. Attempts at photorealistic painting tend to come at a later age. Neural representations of scenes, objects and faces elicited by line drawings were found to be

**Funding:** This work was supported by grants by the Natural Sciences and Engineering Research Council of Canada (RGPIN-2020-04097) and the Social Sciences and Humanities Research Council of Canada (430-2017-01189) to DBW. The funders had no role in study design, data collection and analysis, decision to publish, or preparation of the manuscript.

**Competing interests:** The authors have declared that no competing interests exist.

equivalent to those elicited by the corresponding color photographs in visual cortex [11–13], although the equivalence for scenes appears to be restricted to the level of scene categories, not scene identity [14]. These similarities of line drawings to photo-realistic depiction provide interesting insights into visual brain functions [15].

However, not all line drawings convey visual information equally well. The quality of a drawing, and thereby its usefulness to convey visual content, depends on the experience and on the particular choices of the artist. Lines are placed for different reasons, e.g., to indicate the boundaries of objects, structure within objects, occlusion of one object by another, texture elements, or the outlines of shadows cast by objects [16–19]. We here demonstrate that some lines are more important than others for conveying visual information, and we establish a tight relationship between the importance of contour lines, artistic expertise, and the ability of line drawings to convey scene content. In a series of three experiments, participants with varying degrees of artistic experience were asked to trace the important contours in a set of photographs of complex real-world scenes (Experiment 1). Then, an independent group of participants rank-ordered the drawings belonging to the same photograph, allowing us to establish a measure of contour importance (Experiment 2). Finally, we test the effect of contour importance for scene categorization (Experiment 3).

## Experiment 1: Tracing scene photographs

In this experiment we seek to determine how consistent individuals are at drawing lines as well as effects of artistic expertise on tracing photographs. To this end, we performed a controlled drawing experiment, in which we asked participants with varying degrees of artistic experience to trace the important contours in a set of 18 photographs of real-world scenes. We chose a tracing task rather than free drawing or copying the scenes from the photographs so that we could match the contours in the drawings by different participants based on their spatial location. The task still gives participants freedom to decide which lines they deem important enough to include in their drawings.

### Methods

**Participants.**  We recruited two groups of participants: 14 students majoring in visual arts or architecture from the University of Toronto and the Ontario College of Art and Design University (artists) participated in a two-hour drawing experiment for monetary compensation. In addition, 44 undergraduate students majoring in psychology at the University of Toronto (non-artists) participated in the study for one hour for course credit. The sample size of the artist group was determined by the number of artists who responded to our call for participation. We chose the size of the control sample to approximately equate the total number of drawings produced by the two groups.

We asked all participants to indicate how many years of experience in producing visual arts they had, how many full-year arts courses they had taken, and how much time they typically spend drawing per week. All participants reported normal or corrected-to-normal vision and provided written informed consent. The experiment was approved by the Research Ethics Board of the University of Toronto (Protocol #30999) and followed the guidelines set out in the Declaration of Helsinki.

**Images.**  The 18 images used in this experiment were chosen from a set of photographs of real-world scenes [13]. The images have been rated by an independent group of participants on how well they represent one of six scene categories: beaches, city streets, forests, highways, mountains, and offices. Since knowledge of category membership may aid people in constructing a clearer representation of the scene [20], the top three exemplar photographs from each

scene category were chosen for a total of 18 stimulus images. Four images distinct from the experimental categories were used for practice trials: a regular concave decagon (five-pointed star), a line drawing of a banana, a photograph of a leaf in isolation, and a photograph of penguins with background (Fig 1, left).

**Procedure.** Participants were seated at a computer station, equipped with a cathode ray tube monitor (Dell) as well as a Wacom Cintiq 13HD Interactive Pen Display graphics tablet. The dual-screen experiment was programmed using the Psychophysics Toolbox in Matlab [21], running under the Linux operating system. The CRT computer screen displayed instructions and the original 800x600 pixels photograph as reference during each trial, while the graphics tablet displayed a 1440x1080 pixels copy of the original reference image overlaid with a blank white sheet at 50% transparency for tracing.

Participants received both verbal and written instructions as well as a demonstration on how to use the graphics tablet at the beginning of the experiment. They started each contour line by pressing a button on the side of the graphics pen. They could draw smooth curves as well as set anchor points for quickly making straight edges while holding down the button. The latter function is similar to the Polygonal Lasso Tool in Adobe Photoshop. Traced contours were rendered in black on the semi-transparent overlay. There are 4 control buttons on the non-dominant-hand side of the tablet: NEXT and FINISH for navigating between trials, UNDO for deleting the most recent contour, and SHOW for displaying the drawing without the underlying tracing image (Fig 1). Participants were asked to draw clean outlines of the original image and to avoid extraneous features including repeated lines, shading or labels.

Participants performed four practice trials with stimuli of increasing complexity, intended to familiarize them with the tracing task and the software interface (Fig 1, right). Then, before the experimental trials began, they were given the following instructions:

*For every image, please annotate all important and salient lines, including closed loops (e.g., boundary of a monitor) and open lines (e.g., boundaries of a road). Our requirement is that, by looking only at the annotated line drawings, a human observer can recognize the scene and salient objects within the image.*

During each trial, participants had up to 10 minutes to trace a photograph from the stimulus set. They could proceed at their own pace by finishing a drawing early, or taking a break before the start of the next trial. Non-artists completed 5 trials. Artists were asked to trace as many pictures as they could comfortably do in 2 hours. The 18 stimuli were presented in random order to minimize the influence of practice and fatigue. We recorded spatial coordinates and timing of the strokes for all lines in the final drawings.

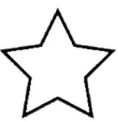 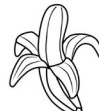 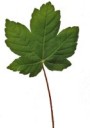 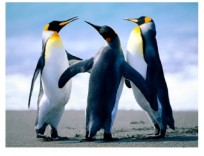 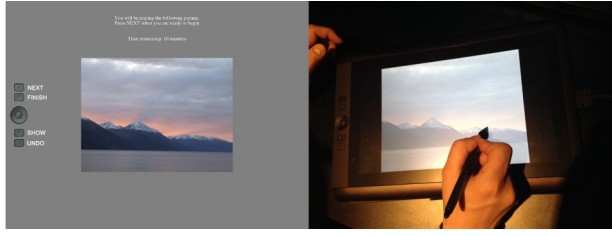

**Fig 1. Setup for Experiment 1.** Left: Images similar to these were use as practice stimuli to familiarize participants with the tracing task and program interface. Images under Creative Commons Licence by Supachoteamorn, Aryadna Pons, Ksanyi, and Christopher Michel. Right Experiment setup with a pen display graphics tablet. The computer screen displayed instructions and the original 800x600 pixels photograph as reference during the trial. The graphics tablet displayed a 1440 x 1080 pixels copy of the image with a semi-transparent overlay for tracing. The Next, Finish, Show, and Undo functions corresponding to each button on the side of the graphics table were displayed during the entire experiment. Images by the authors. The set of 18 photographs are available at: https://osf.io/x9uj5/.

## Results

One artist was excluded from the analysis, because she obviously did not make an effort to produce recognizable drawings. The remaining 13 artists (12 women; age 18–48, mean age 25; 3 left-handed) reported to have 4 to 32 years of arts experience (median 10), had taken 0.5 to 20 full-year arts courses (median 4.5), and draw for 1 to 52 hours per week (median 5). We excluded two non-artists from the analysis–one for adding handwritten words to the drawings, and one for exceedingly sloppy work. The remaining 42 non-artists (26 female, age 17–24 years, mean age 19 years; 2 left-handed) reported up to 12 years of arts experience (median 1) but did not report having taken any arts courses or regularly practicing drawing.

As expected, artists had more years of self-reported art experience, with a median of 10 years compared to 1 year for controls ($t(12) = 5.75$, $p < 0.001$, two-sample t test). The median number of full year-equivalent drawing-related courses taken by artists was 4.5 (range 0.5–18) and they spent a median 5 hours/week practicing drawing (range 1–52). The artist group was older (25.23 versus 19.24 years), more female predominant (92% vs 62%), and less right hand dominant (77% vs 95%).

We collected a total of 194 line drawings from artists and 159 from non-artists. We added the 18 drawings commissioned from trained artists for [13] for a grand total of 371 line drawings, with 15 to 24 (median 21) drawings for each photograph.

## Experiment 2: Ranking line drawings

The quality of the line drawings collected in Experiment 1 varied considerably. For an impartial assessment, we performed a ranking experiment with an independent group of observers. Furthermore, we establish the match of contours between drawings of the same scene. The match allows us to count how frequently a particular contour was drawn across participants in Experiment 1. We use this drawing frequency as a proxy for the importance of contour lines.

### Methods

**Participants.**   We recruited a separate group of 37 undergraduate students of psychology from the University of Toronto. Twenty-five participants spent one hour on the experiment in lieu of partial course credit. Twelve participants volunteered their time and spent less than one hour, performing fewer ranking trials. This sample size was estimated to produce approximately 10 rankings for each of the 18 original images. Participants were unfamiliar with the photographs and line drawings in the study. All participants reported normal or corrected-to-normal vision and provided written informed consent. The experiment was approved by the Research Ethics Board of the University of Toronto (Protocol #30999) and followed the guidelines set out in the Declaration of Helsinki.

**Ranking procedure.**   We printed all 371 line drawings from Experiment 1 on cardstock (15 x 10 cm) and grouped them according to the identity of the original photographs. Each printed drawing was labeled on the back with a unique identifier. We also printed the 18 original color photographs for reference. Participants were asked to sort each set of drawings from most to least representative of the photo by physically arranging the printed drawings on a table and writing down the identifiers in order. To reduce the difficulty of comparing up to 24 drawings at once, raters were instructed to first rank 5 randomly selected drawings, and then insert each additional one into its appropriate location in the list (Fig 2A). Participants proceeded through randomly selected sets of images at their own pace, finishing between 1 and 7 sets (median 6).

**Data analysis.**   After collecting all rankings, each drawing was assigned a normalized rank score, with 100% indicating high rank and 0% low rank with respect to all drawings produced

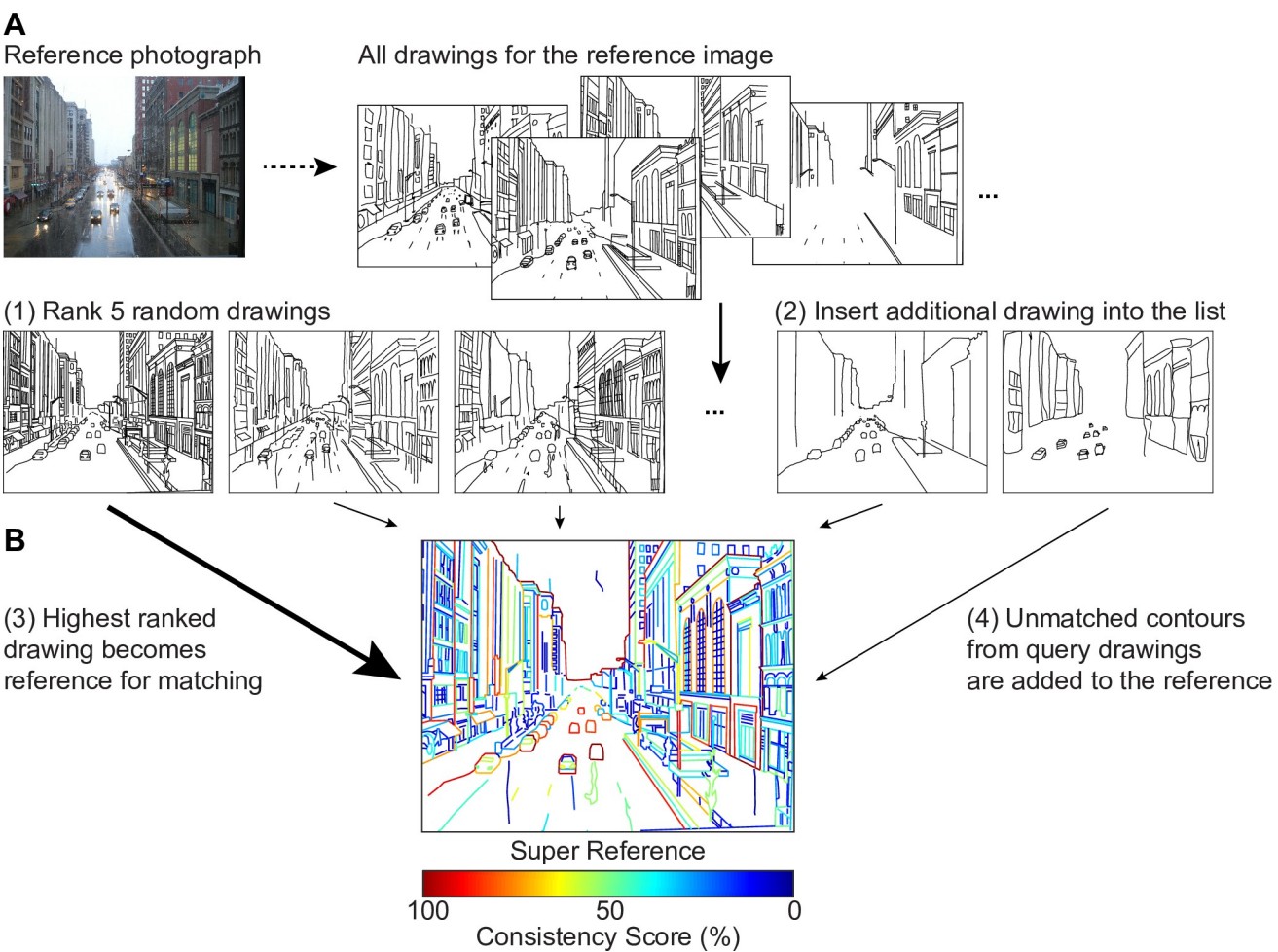

**Fig 2. Procedures for Experiment 2.** (A) Ranking of line drawings was performed by re-arranging images printed on cardstock on a large table. Participants were instructed to first rank five randomly chosen drawings, and to then insert subsequent drawings at the appropriate position. (B) Constructing a super-reference. The highest-ranked drawing served as the initial reference. The other drawings were matched to the reference in ranking orders. Unmatched contours were added to the reference, eventually yielding a super-reference that contains the super set of contours from all drawings. Original photograph under Creative Commons License by Flickr user Cyndy Sims Parr.

for a particular photo. To examine the consistency among raters, we computed Kendall's coefficient of concordance for all rankings of drawings for each image, and performed significance testing using chi-squared statistics. Trials that did not result in a complete ranking of a set of drawings were excluded from the analysis.

For further analysis, we averaged the normalized rank scores for each drawing across raters. The most representative drawing for each image was chosen as the reference for subsequent contour matching between drawings of the same original photograph.

We examined the factors that may have contributed to the ranking with a multiple linear regression analysis. We sought to predict ranking of a particular line drawing based on the artistic experience of its author and the total number of pixels in the drawing, as it is plausible that the amount of detail in a drawing also affects perceived representativeness.

**Contour matching.** To identify which contour lines participants in Experiment 1 most agree on when depicting complex natural scenes, we developed an algorithm to match contour lines. Given two drawings of the same photograph, a human observer can easily identify lines in

each drawing depicting the same element, even though the strokes may vary somewhat in length, number, spatial location and shape. Here we strive to replicate this ability algorithmically.

Let us assume that we have two line drawings, a reference (R) and a query (Q). For each contour in Q we wish to determine the contour in R that is the closest match. In our drawings, contours are composed of a contiguous set of straight line segments. To make the analysis of long line segments easier, we split segments longer than 10 pixels into equal-length shorter segments such that all line segments are at most 10 pixels long.

Next, we calculated the Euclidean distance of a given query line segment to each reference line segment. We use the center point of each line segment for this computation. The reference line segment with the shortest distance is considered a match as long the distance is at most 30 pixels. The identifier of the corresponding reference contour is stored with the query contour segment. Any contour segments more than 30 pixels away from the closest reference contour are considered to be without a match in the reference.

This computation potentially leads to conflicting assignments along the length of the query contour. That is, different segments of the same query contour could be assigned to different reference contours. We resolve such conflicts with a two-stage voting process. First, we implemented a majority vote with a sliding window of 25 segments across the query contour. That is, the reference contour identifier most frequently assigned to the individual line segments within the window along the query contour was assigned to the contour segment at the center of the window. This voting process is analogous to median filtering for noise reduction in image processing [22]. Remember that contour segments are at most 10 pixels long, so the sliding window in this procedure is at most 250 pixels wide.

Second, another global majority vote was then applied to all line segments within the query contour to determine the best matching reference contour. This process usually led to a unique assignment of the reference contour and only rarely led to a tie. When ties occurred, they were broken by determining which of the contending reference contours was closest to the query contour over its entire length. Specifically, we computed the sum of the distances of the individual query line segments to the corresponding reference line segment, weighted by the length of the query line segments. The identifier of the reference contour with the smallest sum (the closest overall distance) was assigned to the query contour.

This matching procedure takes into consideration differences in stroke length, number and spatial location as well as minor variations in shape. This method also takes into account the possibility that one contour in the reference is depicted as more than one contour in a query, but does not account for the reverse.

**Constructing the super-reference.** We used the contour matching algorithm in order to construct a super-reference that contained the superset of all contours drawn by any of the individuals in Experiment 1 for a given photograph. In the first step, we used the highest-ranked drawing as the reference and the second highest drawing as the query and proceeded with the matching algorithm as described above. Any query contours that could not be matched to the reference were added to the reference. Next, the third-highest ranked drawing was used as the query. Any query contours that did not match any of the contours in the updated reference were added to the reference, and so on, until all contours in all drawings were accounted for. As a result, this super-reference captures the superset of all contours, i.e., all non-repeating contours for each scene (Fig 2B).

In a second round of matching, each drawing was matched to the super-reference. In this round, we counted the number of times each contour in the super-reference was matched with a contour in a query drawing. Dividing the count by the number of drawings for this particular image resulted in a normalized measure of the frequency with which a particular contour was depicted. We call this number the consistency score.

## Results

Each set of line drawings was ranked by 5 to 12 participants (median 9). Participants strongly agreed on how much each line drawing represents the original photograph, as evidenced by the significant coefficients of concordance that ranged between 0.57 and 0.92 (Table 1). All 18 super-references are shown in Fig 3, with color of the contours coding for their consistency scores.

We performed a multiple linear regression analysis to predict average normalized rank scores ($S_R$) of individual line drawings based on artistic experience ($X$) of the author of the drawing (measured in number of years of artistic experience) and based on the total number of pixels ($N_P$) in the drawing as a covariate. We excluded four drawings by one outlier artist with 32 years of experience from the analysis. The range of artistic experience without this artist was 0–17 years. We also did not include the drawings originally commissioned for [13], since we had no information about the authors of these drawings.

We found a significant regression equation ($F(3, 345) = 104.8$, $p < 0.001$) with an adjusted $R^2$ of 0.472 (Fig 4). The resulting regression equation is:

$$\hat{S}_R = -3.214 + 2.092 \cdot X + 2.633 \cdot N_P - 0.045 \cdot X \cdot N_P \qquad (1)$$

where artistic experience $X$ is expressed in years and $N_P$ in 1000s of pixels. Artistic experience and total number of pixels were significant predictors of rank with positive coefficients (Table 2). That is, more detailed drawings (with more pixels) and drawings by artists with more experience were ranked more highly. The two factors contributed to the ranking of the drawings independently, as there was no significant interaction.

If individuals show such high agreement on a certain set of contours, do they also draw these lines earlier in the trial? To answer this question, we divided the time when a line was drawn within a trial by the total duration of the trial, resulting in relative timing in the range between zero and one for each of the 353 drawings produced in Experiment 1. Since we did not have timing information for the 18 drawings from [13], we excluded these drawings from

**Table 1. Inter-rater agreement of line drawing rankings.**

| Image ID | Kendall's W | df | p-value |
|---|---|---|---|
| Beach 1 | 0.57 | 18 | $7.62 \cdot 10^{-12}$ |
| Beach 2 | 0.67 | 20 | $6.52 \cdot 10^{-14}$ |
| Beach 3 | 0.60 | 17 | $1.60 \cdot 10^{-10}$ |
| City 1 | 0.82 | 21 | 0 |
| City 2 | 0.88 | 22 | 0 |
| City 3 | 0.85 | 21 | 0 |
| Forest 1 | 0.79 | 14 | 0 |
| Forest 2 | 0.72 | 20 | 0 |
| Forest 3 | 0.64 | 23 | 0 |
| Highway 1 | 0.72 | 20 | 0 |
| Highway 2 | 0.68 | 16 | $4.96 \cdot 10^{-6}$ |
| Highway 2 | 0.69 | 22 | 0 |
| Mountains 1 | 0.92 | 22 | 0 |
| Mountains 2 | 0.87 | 21 | 0 |
| Mountains 3 | 0.67 | 17 | $2.22 \cdot 10^{-16}$ |
| Office 1 | 0.85 | 20 | 0 |
| Office 2 | 0.83 | 19 | 0 |
| Office 3 | 0.81 | 20 | 0 |

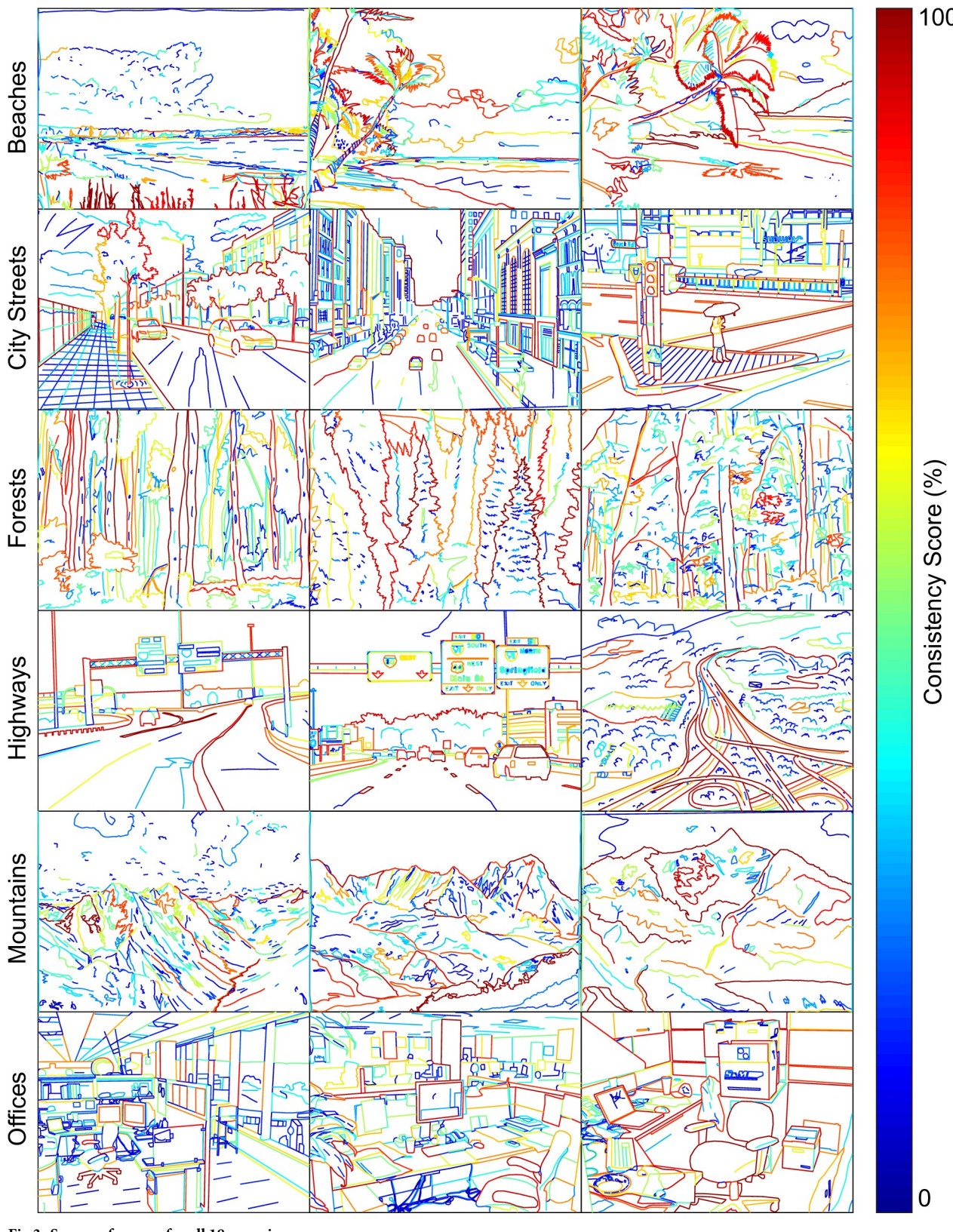

**Fig 3. Super-references for all 18 scene images.**

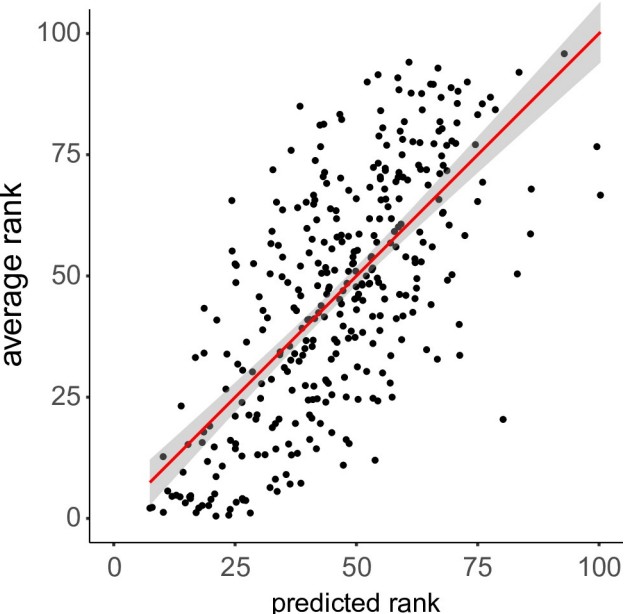

**Fig 4. Average rank of drawings versus the ranks predicted by the multiple linear regression model.** The regression line is shown in red with the 95% confidence interval in gray.

the analysis. Consistency scores of contours in the super-reference drawings were mapped back to the matching contours in the individual drawings.

We fitted a linear mixed-effects model to explain the relationship of consistency scores of individual contours (58480 contours from 353 drawings) with relative timing (fixed effect), with drawing identity as a random-effect covariate. We found a highly significant negative effect of timing ($\beta$ = -0.206; 95% confidence interval = [-0.213,-0.199]; $t(58478)$ = -55.88; $p < 0.001$), with a smaller random effect of image identity ($\beta$ = 0.101; CI = [0.093, 0.109]). This result shows that the most consistent lines tend to be drawn earlier, suggesting that both the consistency and the timing of drawings reflect their importance for perception. We test this hypothesis explicitly in Experiment 3.

## Experiment 3: Effect of line consistency for scene perception

In Experiment 2, we computed the consistency score by matching the drawings of the same photograph by several participants to a super-reference drawing. Does a high consistency score for a particular contour mean that the contour is more important for representing scene content? We address this question with a fast scene categorization experiment with stimuli designed to maximize the difference in consistency scores.

Since contours represent what participants deemed to be important and salient edges in the scene, we hypothesized that the frequency at which a contour is drawn across participants is a

**Table 2. Regression table for the regression in Eq 1.**

|  | coefficient | SE | t-stat | p-value |
|---|---|---|---|---|
| *Intercept* | -3.214 | 3.902 | -0.824 | 0.411 |
| *Artistic Experience* | 2.092 | 0.626 | 3.344 | $9.17 \cdot 10^{-4}$ |
| *Number of Pixels* | 2.633 | 0.246 | 10.719 | $< 2 \cdot 10^{-16}$ |
| *Experience* Pixels* | -0.045 | 0.037 | -1.228 | 0.220 |

good measure of its relative importance for perceiving the scene. To test this hypothesis, we used the consistency scores computed for the super-references in Experiment 2 to construct half-drawings that contained either the most or the least consistent half of the contours. We used these half-drawings as stimuli for a six-alternative forced-choice (6AFC) scene categorization experiment. We hypothesized that the most consistent half-drawings would enable more accurate recognition compared to the least consistent half drawings.

Furthermore, we sought to determine what distinguishes more important contour lines from less important ones. Contours in a drawing can have different physical causes in the real scene. For instance, they can represent depth discontinuities, such as occlusion boundaries, relate to changes in surface curvature, or be part of textures. We here investigate whether any of these physical roles are more important for conveying scene information than others in drawings of complex real-world scenes.

## Methods

**Participants.** Twenty-five undergraduate students of Psychology (ages 17–28, mean 18.9; 16 female) at the University of Toronto participated in the study for partial course credit. This experiment was performed as an add-on for another, similarly structured main experiment. The sample size was determined based on the requirements of the main experiment.

All participants reported normal or corrected-to-normal vision and provided written informed consent. The experiment was approved by the Research Ethics Board of the University of Toronto (Protocol #30999) and followed the guidelines set out in the Declaration of Helsinki.

**Stimuli.** In order to examine the perceptual characteristics of the most-consistently and least-consistently drawn contours, we divided each of the 18 super-references into a high- and a low-consistency half-drawing according to their consistency scores as computed in Experiment 2. The two drawings did not share any contours, and each contained approximately 50% of the total pixels in the original super-reference (Fig 5A). As a result, we obtained 36 half-line drawings, 18 with the most consistent contours (Fig 6A) and 18 with the least consistent contours (Fig 6B).

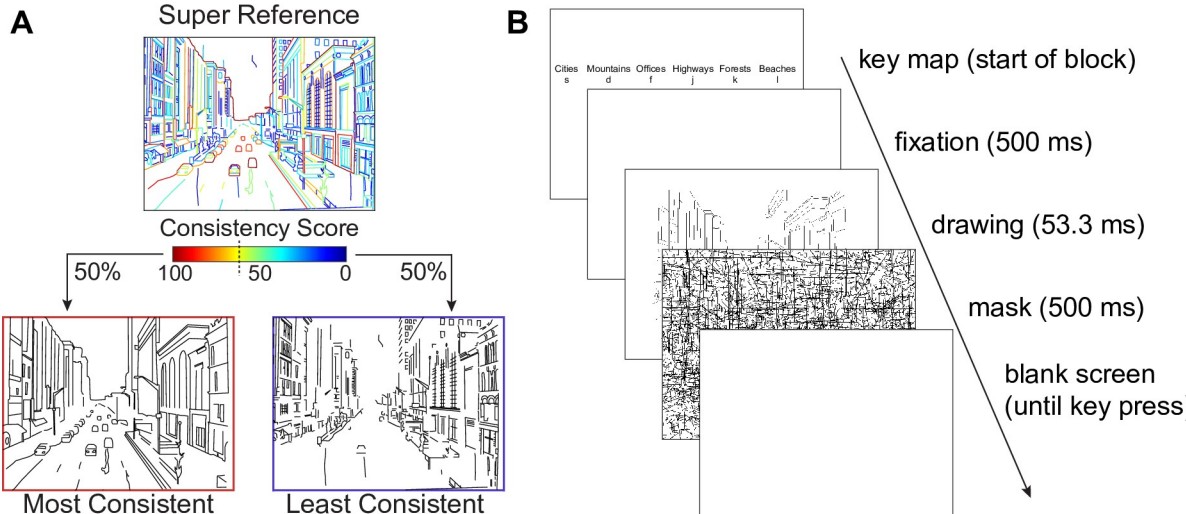

**Fig 5. Design of Experiment 3.** (A) Construction of half drawings according to consistency scores. The two drawings did not share any contours, and each contained approximately 50% of total pixels. (B) Experimental procedure for the six-alternative forced-choice scene categorization experiment.

**A**

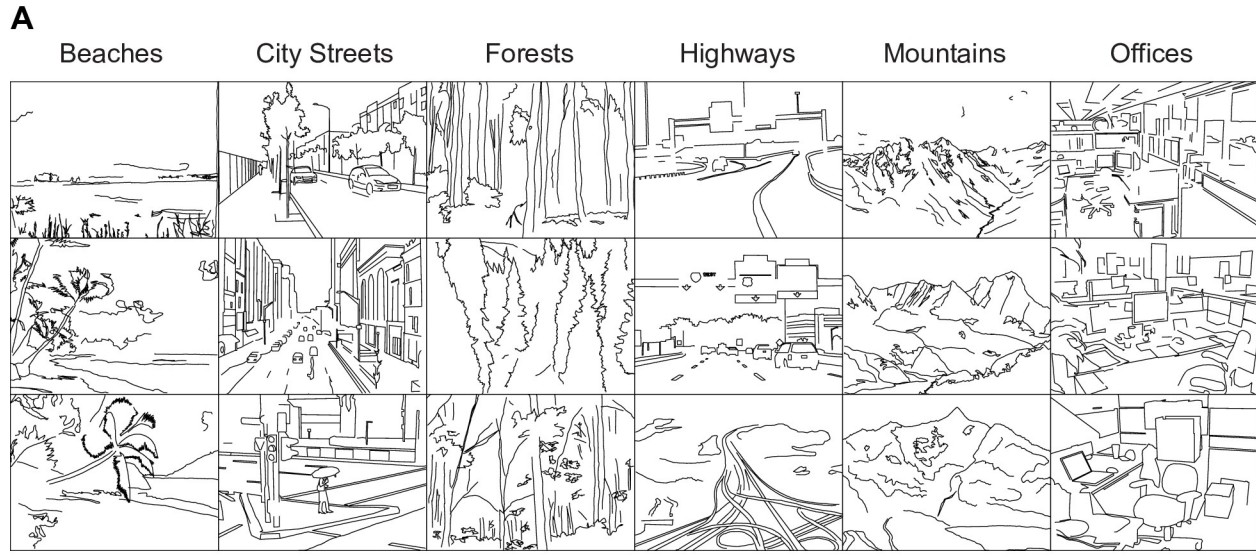

**B**

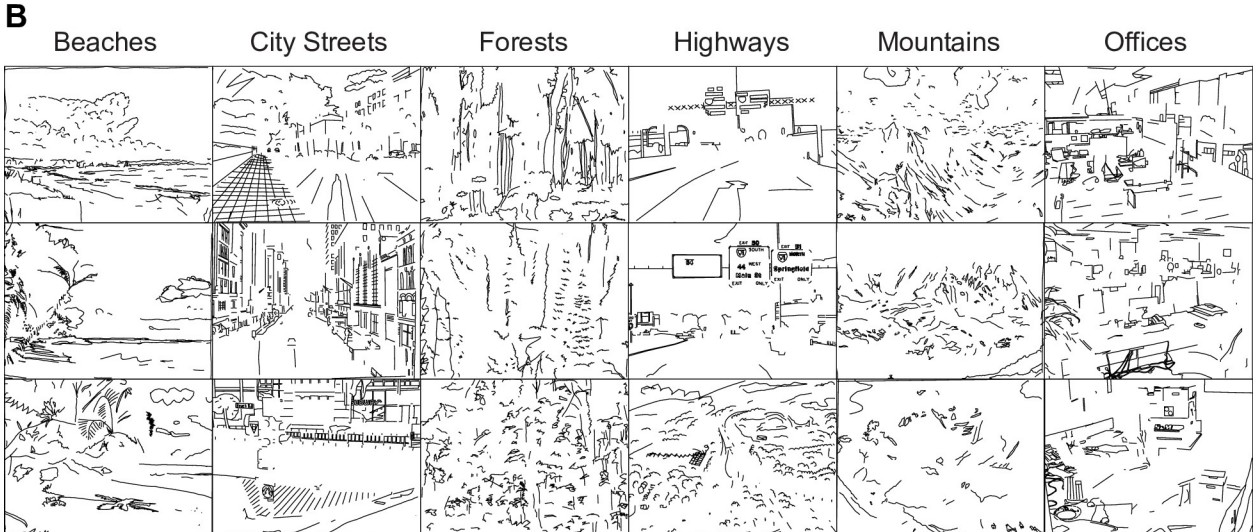

**Fig 6. Stimuli for Experiment 3.** (A) Most consistent half-drawings. (B) Least consistent half-drawings.

**Procedure.** The experiment was performed on a personal computer with Matlab and the Psychophysics Toolbox [21] running on Windows 10. A cathode ray tube monitor (Dell) was used to display the experiment at a resolution of 800x600 pixels with a refresh rate of 150 Hz. Participants were seated approximately 57 cm from the screen.

Participants performed the experiment as an add-on to another, similarly structured main experiment, which will be reported elsewhere. For the main experiment, participants were asked to categorize briefly presented line drawings of real-world scenes into six categories (beaches, city streets, forests, highways, mountains, offices) by pressing one of six buttons on a keyboard (s, d, f for the left hand and j, k, l for the right hand). The assignment of keys to categories was randomized for each participant.

Each trial of the experiment started with a 500 ms fixation period, followed by a brief presentation of the target image, followed by a perceptual mask for 500 ms. The mask consisted of randomly distributed black lines on a white background. After the mask, a blank screen was displayed until a response key was detected (Fig 4B).

In an initial training phase, images were presented for 233 ms (stimulus onset asynchrony, SOA). Participants heard a low tone when they made an error. When participants responded correctly in 17 of the last 18 trials or in 72 trials in total, they moved on to the ramping phase. In the ramping phase, the SOA was linearly decreased from 200 ms to 33 ms over the course of 54 trials. During the subsequent testing phase (360 trials), the SOA was fixed to 53 ms, and participants no longer received feedback.

Following the testing phase for the main experiment, participants performed an additional block of 36 test trials with the half-drawings created for this experiment. Key assignment, SOA, and temporal structure of the experiment remained the same as in the main experiment (see Fig 5). Accuracy of categorizing the drawings was recorded separately for high- and low-consistency drawings and compared using a paired t test.

**Contour types.** To analyze contour types, the first author manually labelled each contour in the super-reference drawings by sequentially overlaying them over the original photograph. She classified contours into four different edge types according to their physical cause [16–18]: texture/albedo edges (change in reflectance across smooth surface), occlusion/depth boundaries (boundaries of objects) [23], surface normal discontinuities (intersecting surfaces, ridges and valleys), and shadow edges (boundary of cast shadows) [19]. For cases where a contour included more than one type of origin, the type that corresponds to the longer portion was chosen. Any contours that could not be assigned clearly to one of these types were labeled as "other". The "other" categories includes "suggestive contours," which occur in locations where surfaces bend away from the observer but do not form a true depth discontinuity [24]. Identifying suggestive contours with certainty occurred too rarely in the real-world complex scenes in this study to justify its own contour category. Once the contours were categorized, we totalled the number of pixels in contours belonging to each type within the most and least consistent half line drawings, and performed significance testing with a fixed-effects two-way ANOVA followed by Bonferroni-corrected paired t tests.

## Results

Participants categorized the half-drawings containing the contours with the lowest consistency scores with 43% accuracy (chance: 16.7%) and the half-drawings with the high-consistency contours with 63% accuracy (Fig 7A). The difference was highly significant (t(14) = 4.21; p < 0.001).

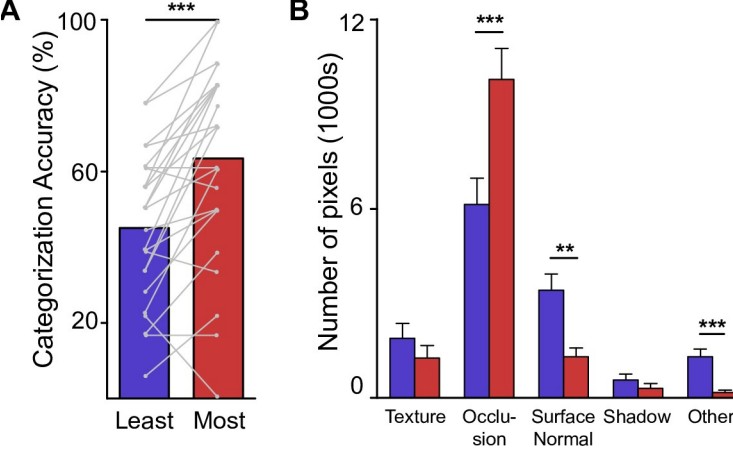

**Fig 7. Results of Experiment 3.** (A) Categorization accuracy for least and most consistent half drawings. (B) Number of pixels in particular types of contours in least and most consistent half drawings. ** p < 0.01, *** p < 0.001.

What makes the most consistently drawn contours so much better at conveying scene information? We annotated all contours in the super references according to their physical cause and analyzed the types separately for the least and the most consistent half drawings (Fig 7B). A 2x5 analysis of variance (ANOVA) with most/least consistent and contour type as factors showed a significant main effect for contour type ($F(4,170) = 79.59$, $p < 0.001$) but not for most/least consistent half drawing ($F(1,170) = 0.014$, $p = 0.906$), as expected since the half drawings are by design equated in the total number of pixels. Importantly, we found a significant interaction ($F(4,170) = 10.98$; $p < 0.001$). The most consistent half drawings contained significantly more pixels belonging to occlusion boundaries, typically object boundaries, than the least consistent half drawings ($t(17) = 8.346$, $p < 0.001$; Bonferroni-corrected for multiple comparisons). This was balanced by more pixels for the least than the most consistent half drawings belonging to contours associated with surface normals ($t(17) = -4.28$, $p = 0.00254$) and contours that could not be clearly assigned ($t(17) = -4.78$, $p < 0.001$). There was no difference in the number of pixels associated with texture boundaries ($t(17) = -1.94$, $p = 0.348$) or cast shadows ($t(17) = -1.65$, $p = 0.588$).

## Discussion

In the three experiments presented here we have established that (1) line drawings are rated as more representative of the depicted scene when they are drawn by experienced artists; (2) the most consistently drawn contours are drawn earlier; (3) drawings containing the most consistently drawn contours are more recognizable than drawings with the least consistently drawn contours; and (4) the most consistent contours are more likely than inconsistent contours to represent occlusion boundaries, that is, the boundaries of objects.

While these results may seem unsurprising in hindsight, this is the first time, to our knowledge, that the ability of humans of various levels of artistic expertise to convey essential information in line drawings has been quantitatively measured for complex real-world scenes. Specifically, we used artistic expertise in the participant populations as a control variable to confirm that artistic experience does indeed result in drawings that are objectively more recognizable. To this end, we established an algorithmic measure of the importance of contours by determining how consistently contours were drawn across all participants. More consistently drawn contours tended to be drawn earlier in the drawing process. Importantly, the fast scene categorization task in Experiment 3 established the perceptual advantage in an unbiased, objective way, since participants in that study were not aware of the authorship of the stimulus drawings. Finally, we found that this perceptual advantage was tied to occlusion boundaries, which represent the shapes of objects in the scenes (Fig 7B).

Recent work on human sketches of individual objects showed similar results for the most agreed-upon contours as well the temporal order of strokes [25]. Interestingly, these experiments included a freehand sketch condition, which resulted in similar profiles of contour agreement as well as temporal order of contours as the tracing condition.

Human proficiency at perceiving objects in complex scenes has been previously linked to edges created from surface normals and depth boundaries in studies with synthetic scenes [16] as well as isolated objects [17, 18, 25] and portraits [26]. Specifically, contours drawn by artists were found to be in good agreement with one another [17], contours were found to be placed at locations optimal for depicting 3D shape and aesthetic quality [18], and artists' drawings were found to define the convex hull area of the depicted objects early on in the drawing process [25]. Our analysis of the roles of contours in Experiment 3 confirms these findings in a real-world scene setting. These findings are consistent with the important role of contour junctions for the perception of objects and scenes, as contour junctions serve as a low-level cue to

spatial relations such as occlusion [27–29]. Moreover, we show how artists prioritize contours that lead to drawings which are more representative of the depicted scene (Eq 1) and that lead to better perception of scene gist (Fig 7A). This finding is likely related to the technique of "blocking-in"–a coarse, block-like outline of the proportions of figures and objects in the initial phase of drawing [30]. This technique may lead trained artists to initially prioritize contours that convey global shape over contours that convey finer details (see also [25]).

While our work involves tracing an image, previous work has shown that when asked to produce a line drawing of a recently viewed scene, non-artists draw many of the object boundaries, and in roughly the correct location [31]. This suggests that humans intuitively know which lines are important to convey the meaning of a scene, but that artistic training improves upon this ability, including improvements in the efficiency in object encoding [18, 32]. Further supporting this notion, aphantasia has been linked specifically to deficits in correctly recollecting objects in a drawing task [33].

Recent computer vision work has demonstrated that non-artists can be trained to prioritize the most important lines and draw them earlier than unimportant lines [34]. Furthermore, even artificial sketch generators, when trained to create a sketch to convey the essence of an image with as few strokes as possible, learn to first draw lines that convey global structure and shape, prior to any details [34]. In fact, there have recently been a number of artificial neural networks trained to generate sketches that are as easily recognizable as those generated by a human [35, 36]. We here show that drawings that take advantage of the visual system's mechanisms for understanding scenes will be more easily interpreted [37].

It is important to point out that we are here not addressing any issues of artistic style, artistic expression or their relationship to perceived aesthetic value. In particular, we make no claims of labeling the line drawings created in this highly controlled study as "artwork." Visual artistic expression, although sometimes concerned with faithful representation of the real world, involves many more aspects, such as composition, emotional content, and frequently metaphorical allusions that transcend the figurative content of the physical artwork. Nevertheless, our findings may help to illuminate what attributes make artwork recognizable, often despite extreme distortions or extreme simplification–for example, gesture drawings or Cubist paintings.

To conclude, we have presented a set of controlled experiments on the production and perception of line drawings for conveying the content of complex real-world scenes. We found that contours drawn most consistently across individuals are most effective at conveying scene content, that contours drawn earlier in the drawing process show higher consistency, and that trained artists are more likely to draw consistent contours. More consistent contours are more likely than less consistent contours to convey occlusion boundaries, which signal the shape of objects in a scene as well as their spatial relationships.

## Acknowledgments

We thank Profs. Natalie Waldburger and Amy Swartz for insightful discussions and for help with recruiting participants from OCAD University.

## Author Contributions

**Conceptualization:** Heping Sheng, Dirk B. Walther.

**Formal analysis:** Heping Sheng, John Wilder, Dirk B. Walther.

**Funding acquisition:** Dirk B. Walther.

**Investigation:** Heping Sheng, John Wilder.

**Methodology:** Heping Sheng, Dirk B. Walther.

**Project administration:** Dirk B. Walther.

**Software:** Heping Sheng.

**Supervision:** Dirk B. Walther.

**Visualization:** Heping Sheng.

**Writing – original draft:** Heping Sheng, Dirk B. Walther.

**Writing – review & editing:** Heping Sheng, John Wilder, Dirk B. Walther.

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
