## [Decision Letter · Decision Letter 0]

6 Sep 2021

PONE-D-21-25287Where to draw the line?PLOS ONE

Dear Dirk,

Thank you for submitting your manuscript to PLOS ONE. After careful consideration, we feel that it has merit but does not fully meet PLOS ONE’s publication criteria as it currently stands. Therefore, we invite you to submit a revised version of the manuscript that addresses the points raised during the review process. Please ensure that your decision is justified on PLOS ONE’s publication criteria and not, for example, on novelty or perceived impact.

I look forward to receiving your revised manuscript.

Best wishes,

Markus 

---

Markus Lappe

Academic Editor

PLOS ONE

Journal Requirements:

3. Please modify the title to ensure that it is meeting PLOS’ guidelines (https://journals.plos.org/plosone/s/submission-guidelines#loc-title). In particular, the title should be "specific, descriptive, concise, and comprehensible to readers outside the field" and in this case it is not informative and specific about your study's scope and methodology.

"We thank Profs. Natalie Waldburger and Amy Swartz for insightful discussions and for help with recruiting participants from OCAD University. This work was supported by the Natural Sciences and Engineering Research Council of Canada (RGPIN-2020-04097) and the Social Sciences and Humanities Research Council of Canada (430-2017-01189)."

"This work was supported by grants by the Natural Sciences and Engineering Research Council of Canada (RGPIN-2020-04097) and the Social Sciences and Humanities Research Council of Canada (430-2017-01189) to DBW. The funders had no role in study design, data collection and analysis, decision to publish, or preparation of the manuscript."

5. We note that Figures 1 & 2 in your submission contain copyrighted images. All PLOS content is published under the Creative Commons Attribution License (CC BY 4.0), which means that the manuscript, images, and Supporting Information files will be freely available online, and any third party is permitted to access, download, copy, distribute, and use these materials in any way, even commercially, with proper attribution. For more information, see our copyright guidelines: http://journals.plos.org/plosone/s/licenses-and-copyright.

a. You may seek permission from the original copyright holder of Figures 1 & 2 to publish the content specifically under the CC BY 4.0 license. 

Reviewers' comments:

Reviewer's Responses to Questions

**Comments to the Author**

1. Is the manuscript technically sound, and do the data support the conclusions?

Reviewer #1: Yes

Reviewer #2: Yes

2. Has the statistical analysis been performed appropriately and rigorously? 

Reviewer #1: I Don't Know

Reviewer #2: Yes

3. Have the authors made all data underlying the findings in their manuscript fully available?

Reviewer #1: Yes

Reviewer #2: Yes

4. Is the manuscript presented in an intelligible fashion and written in standard English?

Reviewer #1: Yes

Reviewer #2: Yes

5. Review Comments to the Author

Reviewer #1: Overall, this paper presents a welcome addition to the literature on line drawing, showing that the most-commonly drawn lines on real scenes correspond to occlusion boundaries, and connecting these lines to artist experience and drawing order. As I am not experienced in publishing this kind of psychological study, I cannot evaluate the details of the experimental setup according to the standards of this field, but, to my eye, it all appears sensible and worthwhile.

My only concern is a lack of citations to highly-relevant studies performed with human artists to address several of these same questions. The submission provides complementary information to these studies, since it operates on a dataset of photographs of real-world scenes, rather than computer-generated imagery of individual objects. There are various pros and cons to the different methodologies. An advantage of the submission is that these are real photographs; a disadvantage of the submission is that the line coding is ad hoc and omits some categories of lines (that may or may not be relevant to these scene categories).

The most immediately-relevant paper was published very recently. This paper analyzes drawing order and uses precise definitions of line styles, and uses photorealistic imagery:

Zeyu Wang, Sherry Qiu, Nicole Feng, Holly Rushmeier, Leonard McMillan, Julie Dorsey

Tracing Versus Freehand for Evaluating Computer-Generated Drawings

ACM Transactions on Graphics (SIGGRAPH), 2021

These older papers are also highly relevant, including analysis of which lines people draw and the order in different cases:

Forrester Cole, Aleksey Golovinskiy, Alex Limpaecher, Heather Stoddart Barros, Adam Finkelstein, Thomas Funkhouser, and Szymon Rusinkiewicz.

"Where Do People Draw Lines?"

ACM Transactions on Graphics 27(3), August 2008.

"How Well Do Line Drawings Depict Shape?," Forrester Cole, Kevin Sanik, Doug DeCarlo, Adam Finkelstein, Thomas Funkhouser, Szymon Rusinkiewicz, and Manish Singh, ACM Transactions on Graphics 28(3)

Berger, I., Shamir, A., Mahler, M., Carter, E., & Hodgins, J. (2013). Style and abstraction in portrait sketching. ACM Transactions on Graphics (TOG), 32(4), 1-12.

Reviewer #2: The manuscript, entitled "Where to draw the line?", by H. Sheng, J. Wilder, and D. B. Walther reports a study testing how people draw line-drawings representing 3D scenes. This is an important and fundamental question in Vision science and the authors addressed this question using a sophisticated method. The study is composed of 3 experiments and they are closely tied to one another. Their line-drawings were generated by people (artists and non-artists) under a controlled condition from photos of 3D scenes in Experiment 1. These drawings were evaluated by different groups of people in Experiments 2 and 3. The drawings were also analyzed by the authors and the results of their analysis were compared with the results of their experiments. The authors found that there were contours that were drawn consistently across the participants (artists and non-artists) in Experiment 1. These consistent contours, which were drawn earlier than the other contours, represented the scenes drawn in the drawings well. These consistent contours often represented the occluding boundaries of objects in the scenes.

The study is very interesting, as well as done very well, and I have only a few minor suggestions.

1) The authors categorize contours the drawings into 4 types (L. 385-388): texture/albedo edges, occlusion/depth boundaries, surface normal discontinuities, and shadow edges. There are, however, some other types of contours that represent 3D information in scenes: e.g. ridges and suggestive-contours. The 3D perception from these two types of contours was tested in Cole et al. (Cole, Sanik, DeCarlo, Finkelstein, Funkhouser, Rusinkiewicz & Singh, 2009, ACM SIGGRAPH).

2) The perception of contours representing shadow edges is discussed in Metzger (1936/2006, Figure 132).

3) L.229-242. This paragraph is unclear and should be revised for clarity.

6. PLOS authors have the option to publish the peer review history of their article (what does this mean?). If published, this will include your full peer review and any attached files.

Reviewer #1: No

Reviewer #2: No

---

## [Author Response · Author response to Decision Letter 0]

22 Sep 2021

Toronto, Sept. 21st, 2021

Dear Prof. Lappe, dear Reviewers

Thank you for your overall positive assessment of our manuscript “Where to draw the line?” We appreciate the input from the two reviewers regarding additional references as well as a more in-depth discussion of contour types. We have revised our manuscript accordingly. Please find our point-by-point response below.

Dirk B. Walther

Reviewer #1: Overall, this paper presents a welcome addition to the literature on line drawing, showing that the most-commonly drawn lines on real scenes correspond to occlusion boundaries, and connecting these lines to artist experience and drawing order. As I am not experienced in publishing this kind of psychological study, I cannot evaluate the details of the experimental setup according to the standards of this field, but, to my eye, it all appears sensible and worthwhile.

We thank the reviewer for the positive feedback and appreciation of our work.

My only concern is a lack of citations to highly-relevant studies performed with human artists to address several of these same questions. The submission provides complementary information to these studies, since it operates on a dataset of photographs of real-world scenes, rather than computer-generated imagery of individual objects. There are various pros and cons to the different methodologies. An advantage of the submission is that these are real photographs; a disadvantage of the submission is that the line coding is ad hoc and omits some categories of lines (that may or may not be relevant to these scene categories).

The most immediately-relevant paper was published very recently. This paper analyzes drawing order and uses precise definitions of line styles, and uses photorealistic imagery:

Zeyu Wang, Sherry Qiu, Nicole Feng, Holly Rushmeier, Leonard McMillan, Julie Dorsey,Tracing Versus Freehand for Evaluating Computer-Generated Drawings, ACM Transactions on Graphics (SIGGRAPH), 2021

These older papers are also highly relevant, including analysis of which lines people draw and the order in different cases:

Forrester Cole, Aleksey Golovinskiy, Alex Limpaecher, Heather Stoddart Barros, Adam Finkelstein, Thomas Funkhouser, and Szymon Rusinkiewicz.

"Where Do People Draw Lines?", ACM Transactions on Graphics 27(3), August 2008.

"How Well Do Line Drawings Depict Shape?," Forrester Cole, Kevin Sanik, Doug DeCarlo, Adam Finkelstein, Thomas Funkhouser, Szymon Rusinkiewicz, and Manish Singh, ACM Transactions on Graphics 28(3)

Berger, I., Shamir, A., Mahler, M., Carter, E., & Hodgins, J. (2013). Style and abstraction in portrait sketching. ACM Transactions on Graphics (TOG), 32(4), 1-12.

We thank the reviewer for making us aware of these publications, especially the recent SIGGRAPH paper. We had neglected to consider the field of computer graphics in our literature search. But as the reviewer points out, these papers are highly relevant! We now include them in the discussion section and specifically position our work with respect to them.

Recent work on human sketches of individual objects showed similar results for the most agreed-upon contours as well the temporal order of strokes (25). Interestingly, these experiments included a freehand sketch condition, which resulted in similar profiles of contour agreement as well as temporal order of contours as the tracing condition. 

Human proﬁciency at perceiving objects in complex scenes has been previously linked to edges created from surface normals and depth boundaries in studies with synthetic scenes (16) as well as isolated objects (17,18,25) and portraits (26). Specifically, contours drawn by artists were found to be in good agreement with one another (17), contours were found to be placed at locations optimal for depicting 3D shape and aesthetic quality (18), and artists’ drawings were found to define the convex hull area of the depicted objects early on in the drawing process (25). Our analysis of the roles of contours in Experiment 3 confirms these findings in a real-world scene setting. These findings are consistent with the important role of contour junctions for the perception of objects and scenes, as contour junctions serve as a low-level cue to spatial relations such as occlusion (27–29). Moreover, we show how artists prioritize contours that lead to drawings which are more representative of the depicted scene (Equation 1) and that lead to better perception of scene gist (Figure 7A). This finding is likely related to the technique of “blocking-in” – a coarse, block-like outline of the proportions of figures and objects in the initial phase of drawing (30). This technique may lead trained artists to initially prioritize contours that convey global shape over contours that convey finer details (see also (25)).

Reviewer #2: The manuscript, entitled "Where to draw the line?", by H. Sheng, J. Wilder, and D. B. Walther reports a study testing how people draw line-drawings representing 3D scenes. This is an important and fundamental question in Vision science and the authors addressed this question using a sophisticated method. The study is composed of 3 experiments and they are closely tied to one another. Their line-drawings were generated by people (artists and non-artists) under a controlled condition from photos of 3D scenes in Experiment 1. These drawings were evaluated by different groups of people in Experiments 2 and 3. The drawings were also analyzed by the authors and the results of their analysis were compared with the results of their experiments. The authors found that there were contours that were drawn consistently across the participants (artists and non-artists) in Experiment 1. These consistent contours, which were drawn earlier than the other contours, represented the scenes drawn in the drawings well. These consistent contours often represented the occluding boundaries of objects in the scenes.

The study is very interesting, as well as done very well, and I have only a few minor suggestions.

We thank the reviewer for the overall positive assessment.

1) The authors categorize contours the drawings into 4 types (L. 385-388): texture/albedo edges, occlusion/depth boundaries, surface normal discontinuities, and shadow edges. There are, however, some other types of contours that represent 3D information in scenes: e.g. ridges and suggestive-contours. The 3D perception from these two types of contours was tested in Cole et al. (Cole, Sanik, DeCarlo, Finkelstein, Funkhouser, Rusinkiewicz & Singh, 2009, ACM SIGGRAPH).

We thank the reviewer for this comment. And thank you for pointing out that we neglected to reference the Cole et al. paper. Regarding the contour types mentioned in that paper, we classified "ridges and valleys" as "surface normals" in our manuscript. Suggestive contours were grouped into "other", since we could not determine a physical reason for the particular contours. We now make this correspondence explicitly clear in the Methods section for Experiment 3: 

To analyze contour types, the first author manually labelled each contour in the super-reference drawings by sequentially overlaying them over the original photograph. She classified contours into four different edge types according to their physical cause (16–18): texture/albedo edges (change in reflectance across smooth surface), occlusion/depth boundaries (boundaries of objects) (23), surface normal discontinuities (intersecting surfaces, ridges and valleys), and shadow edges (boundary of cast shadows) (19). For cases where a contour included more than one type of origin, the type that corresponds to the longer portion was chosen. Any contours that could not be assigned clearly to one of these types were labeled as “other”. The “other” categories includes “suggestive contours,” which occur in locations where surfaces bend away from the observer but do not form a true depth discontinuity (24). Identifying suggestive contours with certainty occurred too rarely in the real-world complex scenes in this study to justify its own contour category. Once the contours were categorized, we totalled the number of pixels in contours belonging to each type within the most and least consistent half line drawings, and performed significance testing with a fixed-effects two-way ANOVA followed by Bonferroni-corrected 

2) The perception of contours representing shadow edges is discussed in Metzger (1936/2006, Figure 132).

Thank you for pointing out this important reference. We have added it to the introduction and the methods section for Experiment 3.

3) L.229-242. This paragraph is unclear and should be revised for clarity.

We have split the paragraph into two parts and added additional explanations to improve clarity. Here is the revised text:

This computation potentially leads to conflicting assignments along the length of the query contour. That is, different segments of the same query contour could be assigned to different reference contours. We resolve such conflicts with a two-stage voting process. First, we implemented a majority vote with a sliding window of 25 segments across the query contour. That is, the reference contour identifier most frequently assigned to the individual line segments within the window along the query contour was assigned to the contour segment at the center of the window. This voting process is analogous to median filtering for noise reduction in image processing (22). Remember that contour segments are at most 10 pixels long, so the sliding window in this procedure is at most 250 pixels wide. 

Second, another global majority vote was then applied to all line segments within the query contour to determine the best matching reference contour. This process usually led to a unique assignment of the reference contour and only rarely led to a tie. When ties occurred, they were broken by determining which of the contending reference contours was closest to the query contour over its entire length. Specifically, we computed the sum of the distances of the individual query line segments to the corresponding reference line segment, weighted by the length of the query line segments. The identifier of the reference contour with the smallest sum (the closest overall distance) was assigned to the query contour.

---

## [Editor Report · Decision Letter 1]

27 Sep 2021

Where to draw the line?

PONE-D-21-25287R1

Dear Dirk,

I am pleased to inform you that your manuscript has been judged scientifically suitable for publication and will be formally accepted for publication once it meets all outstanding technical requirements.

Kind regards,

Markus Lappe

Academic Editor

PLOS ONE
---

## [Editor Report · Acceptance letter]

27 Oct 2021

PONE-D-21-25287R1 

Where to draw the line? 

Dear Dr. Walther:

I'm pleased to inform you that your manuscript has been deemed suitable for publication in PLOS ONE. Congratulations! Your manuscript is now with our production department. 

Kind regards, 

on behalf of

Dr. Markus Lappe 

Academic Editor

PLOS ONE